# Mechanisms for <100> interstitial dislocation loops to diffuse in BCC iron

N. Gao[1,2], Z. W. Yao [3,4✉], G. H. Lu[5], H. Q. Deng [6] & F. Gao [7,8✉]

The mobility of dislocation loops in materials is a principle factor in understanding the mechanical strength, and the evolution of microstructures due to deformation and radiation. In body-centered cubic (BCC) iron, the common belief is that <100> interstitial dislocation loops are immobile once formed. However, using self-adaptive accelerated molecular dynamics (SSAMD), a new diffusion mechanism has been discovered for <100> interstitial dislocation loops. The key aspect of the mechanism is the changing of the habit planes between the {100} plane and the {110} plane, which provides a path for the <100> loops to diffuse one-dimensionally. The migration behavior modeled with SSAMD is further confirmed by in-situ transmission electron microscopy (TEM) measurements, and represents a significant step for understanding the formation of <100> loop walls and the mechanical behavior of BCC Fe under irradiation.

[1] Institute of Frontier and Interdisciplinary Science and Key Laboratory of Particle Physics and Particle Irradiation (MOE), ShanDong University, 266237 QingDao, China. [2] Institute of Modern Physics, Chinese Academy of Sciences, 730000 LanZhou, China. [3] Key Laboratory of Bionic Engineering Ministry of Education, Jilin University, 130022 Changchun, People's Republic of China. [4] Department of Mechanical and Materials Engineering, Queen's University, Kingston, ON K7L3N6, Canada. [5] Department of Physics, Beihang University, 100191 Beijing, China. [6] School of Physics and Electronics, Hunan University, 410082 Changsha, China. [7] Department of Nuclear Engineering and Radiological Sciences, University of Michigan, Ann Arbor, MI 48109, USA. [8] Department of Materials Science and Engineering, University of Michigan, Ann Arbor, MI 48109, USA. ✉email: zhongwen.yao@gmail.com; gaofeium@umich.edu

In metals and many important nonmetallic solids, dislocations are one of key defect structures in understanding the mechanical properties of these materials. Until now, the structures and properties of dislocation lines or loops have been a well-researched area. Among the dislocation loops, the prismatic interstitial dislocation loops (PIDLs) formed by quenching, deformation, and irradiation interactions in body-centered cubic (BCC) iron (Fe) and Fe-based alloys have been studied for decades, because of their critical importance to both the mechanical behavior under normal conditions, and the radiation damage of materials used in fission and fusion reactors[1–11]. From a mechanical viewpoint, the PIDLs can be regarded as hard obstacles, thus the preexisting dislocations would bow around them by the Orowan mechanism. The interaction between loops and dislocations generally influences the plastic deformation process, causing material hardening[8] and low temperature embrittlement[9]. In austenitic steels, the Frank loops irradiated to 7 d.p.a. show a higher relative contribution to the strength of materials[12]. Loops also contribute to irradiation creep, swelling and so on. In addition to irradiation effects, PIDLs are also an area of concern in the development of space aircraft, which is affected by high energy particles in space[4]. PIDLs are harmful defects influencing the safety and lifetime of space aircraft. Thus, to understand the properties of these loops is a hot topic for materials used under extreme environments.

In BCC Fe, there are mainly two types of PIDLs observed along the following Burgers vectors (B): 1/2 <111> and <100>[2–7,10,11]. Based on previous investigations, the following phenomena have been observed for these two types of loops: the first occurs during low temperature irradiations, and diffuses fast with a low-energy barrier and the second is almost immobile after its formation during irradiation. In the past decades, the mobilities of 1/2 <111> loops have been extensively studied[3,7,10,13], but due to its sessile properties, a <100> loop is generally regarded as an obstacle, similar to a precipitate in materials. However, recent experimental results of the <100> loops indicate a mobility[14] and this mobility may have significant impacts on the mechanical properties of BCC metals. Thus, understanding the <100> loop diffusion mechanism is a critical step to further explore the mechanical properties of Fe and Fe-based alloys associated with the formation of <100> dislocation loops by either deformation or radiation. In this work, we explore new diffusion mechanisms of <100> loops by combining a recently developed self-adaptive accelerated molecular dynamics method (SAAMD)[15] and in situ transmission electron microscopy (TEM) measurements.

It is well known that classical MD could provide useful information of <100> loops, but MDs timescale limitation has prevented its application from exploring the diffusion properties of <100> loops, which may take place in seconds or minutes. According to previous studies[5–7,10], the diffusion of a <100> loop is expected to have a high energy barrier, and thus, it is desirable to use SAAMD to simulate such diffusion processes. In order to validate simulation results, in situ TEM observations have also been carried out to compare the diffusion mechanisms and to explore the diffusion and rotation of 1/2 <111> loops in BCC Fe by Arawaka et al.[3,7,16]. Therefore, the combination of SAAMD simulations and in situ TEM observations provides a unique opportunity to explore the diffusion of <100> dislocation loops in Fe.

In this work, through atomic simulations, it is demonstrated that a new diffusion mechanism via one-dimensional (1D) diffusion of <100> loops in BCC Fe. The <100> loop's migration mechanism is significantly different from that of a 1/2 <111> loop. The 1/2 <111> loop diffuses through the correlated migration of the individual self-interstitial atom (SIA) within the loop along the <111> direction, whereas the <100> loop migrates

through changing its habit plane from {100} to different {110} planes under given conditions. In order to confirm the mechanism obtained by SAAMD, an approach by introducing stress from an indenter to push the <100> loop has also been applied to the classical MD method by following the same method of ref. [17]. When the external force is applied, its direction is same as the Burgers vector for a given <100> loop. The details are described in the following text. The migration behavior of a <100> loop based on these simulations show the same result observed in SAAMD simulations.

## Results

**Change of habit plane explored by SAAMD.** The results obtained by the SAAMD simulations confirm for the first time the possibility of <100> loop diffusion in BCC Fe, as shown by the movie (movie-1) in Supplementary Movie 1. In the movie, a <100> loop movies forward and backward along its Burgers vector, and the total diffusion distance is ~4.5 nm within ~2.65 μs. Longer time simulation up to seconds shows similar kinetics. From the dynamics behavior of the dislocation loop, the migration process can be analyzed and the corresponding mechanism is able to be explored. Over the course of the simulations, the diffusion of <100> loops is controlled via the habit plane of the <100> loop changing between {100}, {310}, {210}, and different {110} (e.g., (−110) and (10-1)) atomic planes. Slight differences in the energies of switching habit planes based on the different potentials are shown. The energy difference with a <100> loop on {100}, {310}, and {210} is small, while this difference increases with the loop size on the {110} habit plane. Since local, minimum energy states are explored by SAAMD, the above results indicate that the energy difference between these states is expected to be limited. The change of habit planes has also been observed for 1D diffusion of a single 1/2 <111> loop in BCC Fe between {111}, {110}, and {211} planes[11]. For the 1/2 <111> loop, the intersection of these three planes is a line which has a direction along the <110> direction, while the habit planes of a <100> loop described above intersect at a point. The difference of intersecting at a point versus a line are compared to understand the special diffusion of the <100> loop.

The energy difference of the <100> loop on different habit planes, {100} and {110}, with the same Burgers vector, is calculated. For a <100> loop, the Burgers vector always remains the same, but the habit planes can be changed, as described above, where the main difference between these panes is the atomic sequence of the habit planes, i.e., "AB", "ABC", and "ABCD" sequence in the {100}, {130}, and {120} habit planes, respectively, but only "A" sequence in the {110} plane. The examples of the {100} and {110} planes are shown in Fig. 1. The dislocation properties in Fig. 1 are identified by the DXA method[18]. In this work, the energy states of a <100> loop containing various numbers of SIAs located on the {100} and {110} planes are calculated by the following equation:

$$E_l(\text{SIA}) = \frac{E_{\text{tot}}(\text{loop}) - (N + m)E_f^p}{m}, \tag{1}$$

where $E_l(\text{SIA})$ is the formation energy of a loop per SIA. $E_{\text{tot}}(\text{loop})$ is the total energy of a system containing a loop, in which there are $m$ SIAs and the total number of atoms in the system is $N + m$. $E_f^p$ is the energy per atom in a perfect crystal. The energies are calculated after relaxation by the molecular statics (MS) method.

With Eq. (1), the formation energy of a loop located on the {100} and {110} planes is calculated and shown in Table 1: (1) with increasing number of SIAs in the loop, the loop formation energy decreases accordingly, (2) the <100> loop located on the

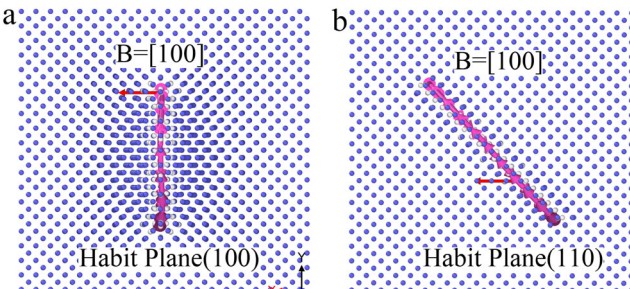

**Fig. 1 Configuration of a <100> loop with its Burgers vector [100] located at different habit planes. a** The (100) and **b** the (110) planes, respectively. In this case, the number of SIAs in the loop is 137. The loop is shown by the pink curve with arrows. The red arrows indicate the Burgers vector of the loop.

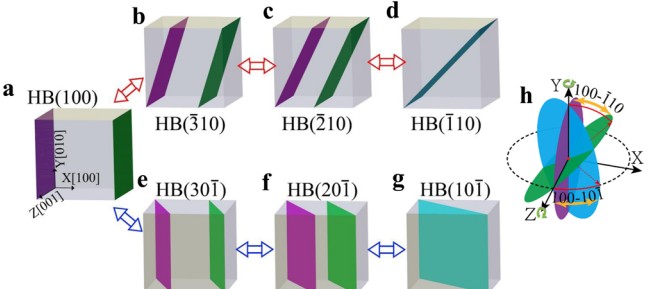

**Fig. 2 Schematic of diffusion mechanisms of a <100> loop on the (100) habit plane with b = [100].** Its habit plane can change from the **a** (100) to the **b** ($\bar{3}$10), the **c** ($\bar{2}$10), and the **d** ($\bar{1}$10) (path 1) or to the **e** (30$\bar{1}$), the **f** (20$\bar{1}$), and the **g** (10$\bar{1}$) (path 2). The schematic of two paths and the cross point are shown on the right side **h**. The different habit planes are marked by the purple, green, and blue, respectively.

**Table 1 Energy states of [100] loops located on habit planes (HB) of the (100) and the (110), as a function of number of SIAs ($N_{SIA}$) in the loops.**

| $N_{SIA}$ | HB(100) | HB(110) |
|---|---|---|
| 69 | 1.1816(±0.0001) | 1.1912(±0.0001) |
| 97 | 1.0405(±0.0001) | 1.0569(±0.0001) |
| 109 | 1.0064(±0.0001) | 1.0133(±0.0001) |
| 137 | 0.8952(±0.0001) | 0.9389(±0.0001) |
| 145 | 0.8834(±0.0001) | 0.9114(±0.0001) |
| 177 | 0.8030(±0.0001) | 0.8479(±0.0001) |

{100} habit plane has lower formation energy than on the {110} plane, and (3) the energy difference of each atom on these two habit planes is very small. It should be noted that with increase in the number of the SIAs in a loop, the computational box is also increased accordingly, from the initial 20–50 unit cells (for 171-SIA loop with radius ~1.5 nm) along each direction to avoid the possible size effect. These results are consistent with the SAAMD results, i.e., a <100> loop can change its habit planes. From a thermodynamic viewpoint, these two states are expected to transfer to one another, as the energy barrier between them can be overcome by temperature fluctuations or by an external stress, as indicated by Eq. (1). The states of the loop on the {130} and the {120} are considered as the intermediate states for the loop to diffuse between the {100} and the {110} habit planes. Therefore, the diffusion behavior of a <100> loop is accompanied by the changing of its habit planes between different states, resulting in its 1D diffusion.

Based on these results, a possible diffusion mechanism of a <100> loop in BCC Fe is obtained. Figure 2 shows that there are at least two paths for a <100> loop to diffuse one Burgers vector distance from the habit plane {100} to {110} via the {130} and the {120} planes. The first path consists of the <100> loop changing its habit plane from the initial (100) plane to (−310), (−210), and then to the (−110) maintaining its [100] Burgers vector, as shown from (a) to (d) by the red planes in Fig. 2. The second path is the <100> loop changes its habit plane from the initial (100) plane to the (30-1), the (20-1), and then to the (10-1) with the same Burgers vector, as shown from (a) to (g) by the red planes in Fig. 2. If the loop jumps between (a) and (b), (b) and (c) or (between (a) and (e), (e) and (f)) the loop only moves its center of mass a distance of 1/6, 1/3, or half of the Burgers vector. If the state changes from (a) to (d) through the path marked by the red planes and from (d) to (a) through the path marked by the green planes (or from (a) to (g) by the red planes and (g) to (a) by the

green planes), the <100> loop moves a distance of one Burgers vector. Different from the diffusion of a 1/2 <111> loop, which can only rotate its habit planes around a line, the diffusion of a <100> loop can migrate along two paths which are connected continuously through the {100} habit planes, i.e., a <100> loop can rotate its habit planes in two ways. From the present simulations, we do not observe a direct path between (b) and (e), (c) and (f), (d) and (g). All these jumping processes can be also observed in the movie-1 provided in Supplementary Movie 1.

During the rotation of the loop between the {100} and {110} planes, there exists an intermediate state by mixing the {100}, {130}, {120}, and {110} habit planes, as shown in Fig. 3a. It should be noted that in the simulations above, the diameter of the <100> loop is up to 2.0 nm. When the loop size increases, the intermediate states appear more frequently. The reason for this phenomenon may be associated with the small energy difference between these habit planes. The connection of these different habit planes ensures that the larger <100> loops between different states jump continuously, such that the minimum energy barrier for their diffusion in 1D can be explored.

The diffusion mechanism of the <100> loop is different from the 1/2 <111> loop, particularly relating to the intersection of these habit planes. As the habit plane of the 1/2 <111> loop intersects at a line, whereas the <100> loop intersects at a point. Thus, the degree of freedom of these two loops during their diffusion is different. The related atomic arrangement during the diffusion is also different, which results in the different stress or strain field during the diffusion. The possible effects of these differences on the radiation damages have been discussed in the following section. When the intersection is a line, it can be regarded as the rotation axis of different habit planes. For a 1/2 <111> loop, the rotation angle from the {111} to the {112} plane is ~19.5°, and 35.3° from the {111} to the {110} plane. It should be noted that the {111} plane is between the {112} and the {110} planes, and thus, with the same rotation axis, it is impossible for a 1/2 <111> loop to rotate its habit plane directly from the {110} to the {112} plane. Instead, the 1/2 <111> loop changes its habit plane from the {112} to the {111}, and then from the {111} to the {110}[11]. The energy barrier for diffusion between these habit planes is low, as indicated by its fast diffusion. For a <100> loop, if its rotation follows one of the above two paths, the rotation angle is ~18.4° from the {100} to the {130}, 8.2° from the {130} to the {120}, and 18.4° from the {120} to the {110}. The energy barriers calculated using NEB method for the <100> loop with a diameter of 1.6 nm are shown in Fig. 4. It is clear that the loop has an energy barrier between 0.9 and 1.2 eV from the {100} through

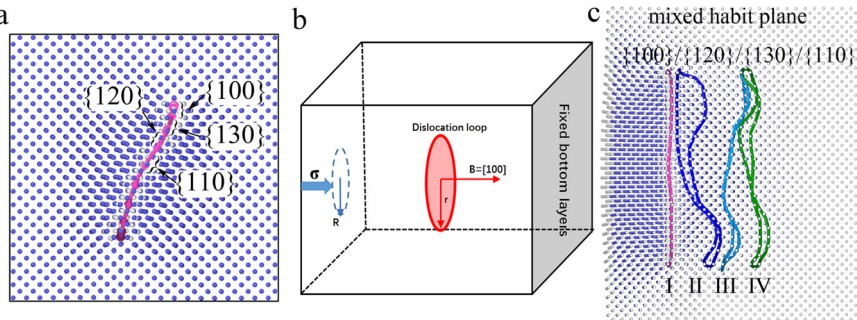

**Fig. 3 Habit planes in SAAMD and nanoindentation simulations. a** The mixed {100}, {130}, {120}, and {110} habit planes of a <100> loop (shown by the pink curve) during its diffusion in SAAMD simulations. **b** Schematic of external stress applied on a <100> loop in BCC Fe with nanoindentation method. The loop plane and Burgers vector are shown in the figure. The evolution of the loop with simulation time is shown in **c** from state I to state IV, shown by different colors.

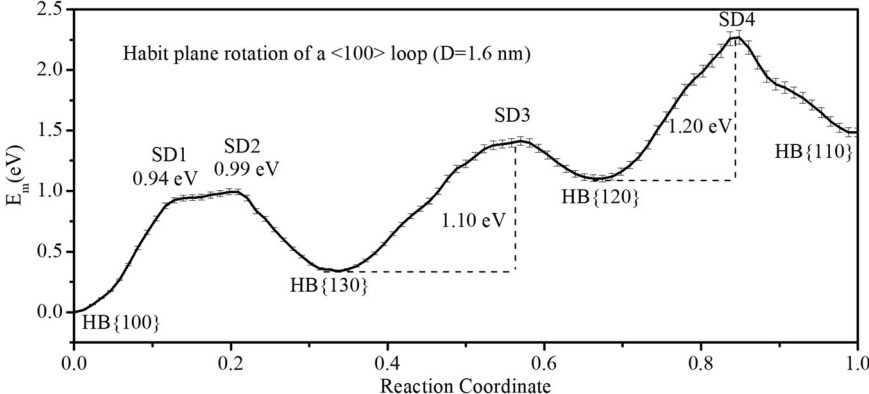

**Fig. 4 Energy barrier for the diffusion of a <100> loop between different habit planes.** The diameter of loop is 1.6 nm, which rotates between the {100}, the {130}, the {120}, and the {110} habit planes. For clarity, the configurations of saddle point (SD) states, SD1–SD4, are shown in Supplementary Fig. 2.

the {130}, the {120} to the {110} habit planes. When the size of the loop increases, the energy barrier is expected to increase. For example, a large loop with a diameter up to 3.0 nm, because of the mixed habit planes, the barrier is between 3.0 and 4.0 eV for the three jumps. Figure 4 shows that there may be other intermediate states with different habit planes instead of the following planes: {100}, {130}, {120}, and {110}. The configurations of saddle point states between above habit planes are shown in Supplementary Fig. 2. From the lattice structure, these habit planes can all be defined as {1k0} planes. For example, the {150} and {170} habit planes with atomic sequence "ABCDE" and "ABCDEFG" are recognized as intermediate states. The present simulations also indicate that the planes closer to the {100} habit plane, i.e., by increasing "k", is prohibited with k >7 for nanoscale <100> loops. These loops are unstable and transfer to the {100} habit plane or the {170} habit plane. Meanwhile, when the size of loop is large enough, for example, >5 nm in diameter, these states may be possible because of the weaker elastic interaction between the segments of loop cores, which is close to the properties of edge dislocations. For the above cases, the high energy barrier still exists and may be one of the possible reasons why it is difficult to observe the diffusion of a <100> loop in BCC Fe, especially for larger loops. However, if the temperature is high enough and the size of loops is small, the diffusion of these <100> loops is expected, resulting in different defect evolution during radiation damage.

Based on the above analysis, it may be difficult to observe the diffusion of a <100> loop only by increasing the temperatures in classical MD simulations because of its time limitation. According to the jump frequency theory described by Arrhenius equation, in

addition to the temperature, the external stress may also provide another possible way to facilitate the state transformation on the energy surface of the system. Hence, in order to confirm the diffusion mechanism obtained from SAAMD simulations, the diffusion of a <100> loop is simulated under the stress along its Burgers vector. The model is shown in Fig. 3b, which is similar to the MD simulation of a nanoindentation. The interaction between the indenter and the box can be described by a strongly repulsive potential. Thus, a cylindrical indenter is applied to the system. In order to avoid the effect induced by a shockwave, which occurs from the fast indentation process in MD simulations, a relaxation process during the indentation is used. After the indenter heads $\delta d$, the indenter remains at the current position, but the system is relaxed for $\delta t$ before the indenter is allowed to head another $\delta d$. In this work, $\delta d = 0.1$ Å and $\delta t = 100$ ps with a time step of 1 fs. The radius of the indenter is 3.0 nm and the radius of the loop is ~3.3 nm (containing 697 SIAs), which is much larger than the loop studied by SAAMD in the above simulations. Without the external stress, the loop remains immobile within the classical MD approach, even with simulation temperatures up to 1000 K. After applying an external stress through the indenter, the diffusion of such large loops is observed at room temperature, as shown in Fig. 3c with the loop positions from I, to II, to III, and to IV as a function of simulation time. After carefully analyzing the states of the loop, especially for the habit planes during its diffusion under the external stress, the change of habit planes is observed to occur along the {100}, the {130}, the {120}, and the {110} planes. The mixed habit plane of the loop is monitored as shown in Fig. 3c. These results are the same as those from the SAAMD simulations. The similar results

are observed with different radii of the indenter and the different sizes of the loops. The external stress provides a way to decrease the energy barrier for a <100> loop to diffuse. The diffusion of a <100> loop through changing its habit planes between the {100}, the {130}, the {120}, and the {110} planes is confirmed in both MD and SAAMD, representing a key diffusion mechanism in BCC Fe.

**Comparison with in situ irradiation and TEM measurements**. In order to validate the diffusion mechanism explored by SAAMD simulations, in situ ion irradiation experiments are performed. The video (movie-2) provided in Supplementary Movie 2 shows direct evidence of the diffusion process of a <100> loop in the single-crystal Fe at 773 K irradiated with 150 keV Fe$^+$ ions. It should be noted that although it is difficult to compare the kinetic process of each atom within the loop, it is still useful and helpful to compare the atomistic simulation results with the in situ TEM observations at 773 K, which can be used to explore the main features of the loop, i.e., the rotation of its habit planes driving its 1D diffusion. The video clearly shows the rotation of the loop from its habit plane of {100} to the other habit planes. Combining with the simulation results and crystal geometry, the process from {130}, to the {120}, and finally to the {110}, is possible. The loop also combines with another <100> loop to form a larger loop, which continues to diffuse via mixed habit planes. These processes are the same as process modeled using SAAMD. The TEM images related with these processes are depicted in Fig. 5. Also, the SAAMD simulations at 600 K present the same results to those observed at the lower temperature; changing habit planes of a <100> loop during its 1D diffusion process. From the recent experimental results[13], the diffusion of a <100> loop observed by the TEM observations can be then used to compare with the present simulation results. Although the internal stress landscape from the atomic simulations may be different from that in the experimental sample, the results from both methods indicate the rotation of the habit plane of a <100> dislocation is a one of the major mechanisms for its diffusion. Thus, the present work is not to focus on which way the loop is triggered to diffuse along its Burgers vector, but instead on the possible diffusion mechanisms of the loop, which has not been explored previously. The reason for above difference for internal stress may be from: (1) uncertainty of energetics of all defects at higher temperatures induced by the empirical potential; until now there is no one empirical potential that can be applied for exactly describing the high temperature elastic properties of BCC Fe, which is vital to obtaining the correct energy landscape of defects at high temperature; (2) the sample surface effect, applying additional stress field on the loop, may be different from the state

obtained with periodic boundary condition used in the simulations. In both the video and TEM images shown in Fig. 5, there are also immobile <100> loops. During the whole process, these loops rotate a small angle, ~5°–10°, from their initial habit planes, without rotation between the {100} plane and the {110} plane. According to the results shown in Fig. 2, such small rotation is not enough for a loop to diffuse, thus, showing immobile properties. In in situ experiments, the motion of a <100> dislocation loop is driven by the Peach-Koehler force generated by other sources, such as surrounding dislocations and irradiated defects. However, if this force is not large enough, the rotation of their habit planes does not occur, and thus, some dislocation loops remain immobile. Except the evidence shown above, in situ TEM observation also shows an indirect evidence that a <100> loop near the surface diffuses along its Burgers vector. According to the TEM diffraction contrast theory, if the habit plane of a <100> loop remains the same, the brightness of the loop should be always the same. However, Fig. 6 clearly demonstrates that the brightness of the loop changes during its diffusion to the near surface. Considering the direct evidence shown in Fig. 5, it can be deduced that the loop takes the different habit planes, and rotates between the {100}, the {130}, the {120}, and the {110} for its diffusion and finally absorption at the surface, shown in Fig. 6. The video (movie-3) provided in Supplementary Movie 3 shows that a <100> loop diffuses along its Burgers vector, and eventually disappears from the surface. In order to confirm these habit planes, we further simulated the TEM images with the TEMACI program developed by Zhou in Oxford[19–21] and compared the simulated results with the images from movie-3, as shown in Fig. 7. In the TEMACI image simulations, the zone axis is taken as [100], diffraction vector is [011], and habit planes are set to be (100), (−310), (−210), and (−110). Accelerating voltage is 200 kV. The results shown in Fig. 7 confirm that the habit planes during the rotation of a <100> loop progressively change from (100) to (−310), (−210), and (−110), which are same as the SAAMD predictions. Thus, both methods explore the diffusion of a <100> loop through the rotation between the {100}, the {130}, the {120}, and the {110} habit planes. The simulation results are validated by experimental observations.

**3D diffusion of <100> loops and effects on radiation damages**. A remaining question is whether the <100> loop could diffuse three-dimentionally in BCC Fe. The diffusion of a <100> loop is tested at higher temperatures, up to 800 K. In addition to the 1D diffusion process explored above, the transformation from the <100> loop to a 1/2 <111> loop is observed, but without direct transformation between two different <100> loops. Transformation between a 1/2 <111> loop and a <100> loop has also been

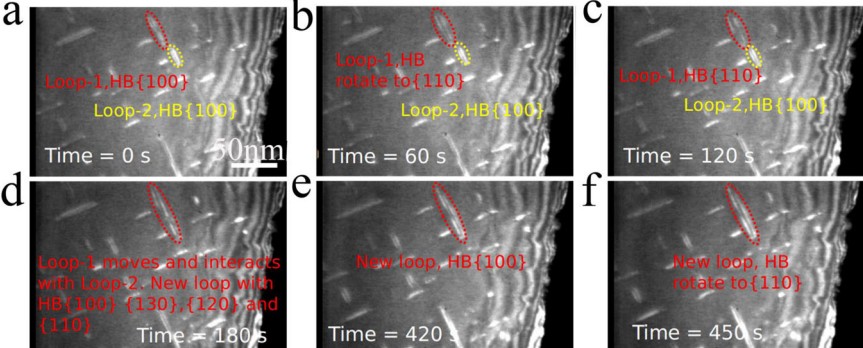

**Fig. 5 Direct in situ TEM observations of a <100> loop diffusion through habit plane rotation.** The rotation of a loop (loop-I) is shown in **a–c**, and two loops are shown by small red and yellow dot ellipses, respectively. The rotation of the large loop after interaction is shown in **d–f**. Total time of in situ observation is up to 450 s.

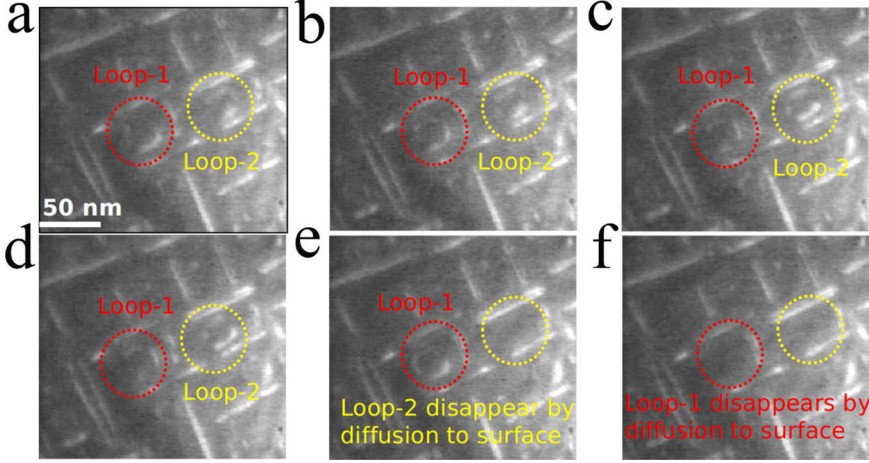

**Fig. 6 Indirect evidence of <100> loop diffusion through the change of their habit planes. a–f** The rotation of habit planes between the {100} and the {110} during their diffusion to the surface along the Burgers vector. Two loops are shown by small red and yellow dot circles, respectively.

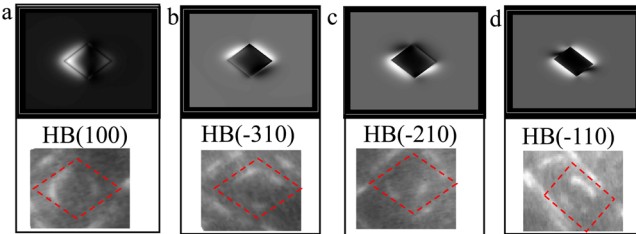

**Fig. 7 Comparison between the simulated TEM images and in situ images for a <100> loop.** The loop is located on different habit planes: **a** (100), **b** (-310), **c** (-210) and **d** (-110). The top images are results simulated by TEMACI program[19–21], and the bottom images are results taken from the in situ observation from Supplementary Movie 3. The shape of loop is marked by the red dot rhombus and rectangle.

observed in previous studies[11]. Since the transformation between different nano-size 1/2 <111> loops are possible[3,7], it is likely that the <100> loop can diffuse in 3D through the state of a 1/2 <111> loop in BCC Fe. Although such a probability for three-step transformation is low, it does provide a possible way for a <100> loop to diffuse in 3D. Therefore, the diffusion of <100> loops provides the key step to form the 1D loop wall in BCC Fe and its alloys[13]. Based on the above results, that under given conditions (e.g., temperature and stress field), the 1D and 3D diffusions of <100> loops may enhance the formation of a superlattice or alignment of <100> loops or 1/2 <111> loops[14,22,23].

As mentioned, the formation of <100> loops has been observed in different BCC metals and alloys, which influences the mechanical properties of materials under extreme conditions (high irradiation field, high temperature, and high stress). If these <100> loops are assumed to be immobile, showing sessile properties, they can be treated as local precipitates and act as pining centers to hinder dislocation motion, resulting in radiation hardening. However, based on the current study, if the diffusion property of a <100> loop is taken into account, especially the intersection of its habit planes as a point during its diffusion, some important effects of <100> loops on radiation damage need to be reconsidered: (1) high efficiency for <100> loops to absorb the point defects and small defect clusters due to the stress field variation during their diffusion, resulting in the quick loop growth (as indicated by Fig. 6 in ref. [14]), and increasing the pinning effect and, thus radiation hardening; (2) formation of local high stress regions, e.g., the formation of loop walls as observed in TEM[14], resulting in high stress concentrations and

related micro-cracks under the effect of both the external temperature and stress fields; (3) similar to 1/2 <111> loops, leading to additional influences on radiation creep and swelling, as explained in the ref. [24]. Therefore, to understand the diffusion mechanism of <100> loops in BCC Fe and BCC Fe-based alloys provides a rich knowledge for understanding radiation damage, which would be helpful in predicting the lifetime of materials under extreme conditions.

## Discussion

In summary, the 1D diffusion mechanism of a <100> loop in BCC Fe has been explored through SAAMD and in situ TEM ion irradiation experiments. The results from both methods exhibit the same conclusion that the change of habit planes between the {100}, the {130}, the {120}, and the {110} is a predominant path for a <100> loop to diffuse in 1D, which provides a key step in understanding the formation of <100> loop walls observed in Fe–Cr alloys[14]. The present study also advances the current understanding of the interaction between <100> loops and dislocation lines, resulting in radiation hardening, micro-crack formation, creep, and swelling. However, it is important to include the diffusion mechanism of <100> loops in microscale or macroscale modeling, such as rate theory, for understanding the fundamentals of radiation damage. These results also illustrate that the mechanisms explored by coupling simulations with experimental observations are able to provide a comprehensive understanding of loop dynamics and more insights into engineering microstructures via ion implantation, high energy particle bombardment, or cold work.

## Methods

**Self-adaptive accelerated molecular dynamics simulations**. The SAAMD simulations are carried out based on the algorithm developed in ref. [15]. The Fe interaction is described by the Ackland Fe potential[25]. However, other potentials, e.g., Mendelev[26] and the "magnetic" Dudarev–Derlet potential[27], have been tested to corroborate the results. These potentials all produced similar results. The directions of computational box are along the [100], [010], and [001] directions, each of which consists of 20 unit cells (containing 16,000 atoms in the simulation box). A <100> loop is located at the center of the box by inserting two extra atomic layers on the (100) plane along the Burgers vector direction without introducing the stacking fault. The radius of such a loop is ~1.0 nm. The total number of atoms in this loop is ~69, which is 0.4% of the total number of atoms in the simulation box. Since the loop is located at the center of the box, the distance between the loop and its image is ~5.7 nm along the $X$ direction, and ~3.7 nm along the $Y$ and $Z$ directions because of the periodic boundary conditions. It should be noted that the strongest interaction between loops is along its Burgers vector direction, but decreases as a function of the reciprocal distance ($\sim 1/r^3$), much faster than an edge dislocation ($\sim 1/r$)[28]. The formation energy of this loop calculated as a function of

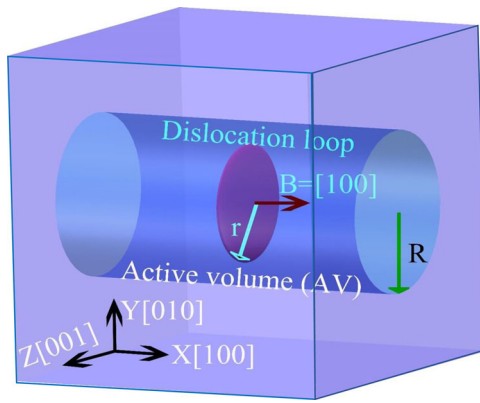

**Fig. 8 Schematic of the cylindrical active volume (AV) selected for SAAMD simulations.** $r$ and $R$ are radius of the <100> loop and AV, respectively.

unit cells has also indicated that 17–18 unit cells along each direction are enough to avoid the image interaction, as also shown in Supplementary Fig. 1. Furthermore, the theory suggested by Cai et al.[29]. has also been used to prove the box with 20 unit cells along each direction in the present work is large enough to avoid the image force from the boundary condition, as shown in Supplementary Figs. 5 and 6. The system is firstly relaxed by the MS method, followed by MD relaxation at 300 K with the Parrinello–Rahmann method[30] to release the stress under NPT ensemble. After full relaxation, the SAAMD is then applied to simulate the migration of <100> dislocation loops. The related calculations with theory developed by Cai et al.[31,32] have also been used to calculate the ratio of image energy to the total energy of system with 20 unit cells containing a loop with radius of 1.0 nm. The calculated result is ~$3.6 \times 10^{-6}$, which could be thus neglected in the present study. The details of these equations and calculations have been provided in Supplementary Note 1.

The SAAMD was developed to model the infrequent atomic scale events, especially those events that occur on a rugged free-energy surface, for example, the diffusion of complexed radiation defects in materials. The boost potential concept suggested by Voter[33] is used to accelerate the dynamics, but our approach is different from the original method by slowly filling the potential valley. However, the key of this filling potential to slowly boost dynamics does not to alter the energy barrier, but just increasing the probability for the event to occur by overcoming the high energy barrier. This approach is self-evolving, where the total displacement of the system at a given temperature is used to construct a boost potential. The method can be used to study not only the migration of atoms on the free surface, but also the migration of helium-vacancy cluster in materials, which has been confirmed by LANL researchers with parallel trajectory splicing (ParSplice) that is a long-time dynamics method developed by Perez et al.[34]. Thus, considering the possible high energy barrier for the diffusion of a <100> loop, the SAAMD is applied for this work.

To use the SAAMD method, the active volume (AV) should be selected first. In the present work, considering the possible migration of a <100> loop along its Burger vector, a cylindrical AV is selected around the <100> loop with a radius 20% larger than that of the loop for the SAAMD simulation, within which all the atoms are accelerated, as shown in Fig. 8. Following the ref. [15], the criteria to determine the state change of the <100> loop is introduced by controlling the maximum displacement of an atom ($D_{max}$) in the AV. The maximum displacement is set to 2.0 Å (~80% of Burgers vector), determined by multiple pre-simulations, for the present study. Once the maximum displacement is reached, the boost potential is removed and classical MD is then used for another 20 ps simulation with a time step of 1 fs, after which, MS is applied for the system to reach a new local stable configuration. Then, the SAAMD is applied again to accelerate the atoms in the AV. The temperature for the SAAMD simulations is 300 K. The details to determine the parameters used in the present SAAMD simulations are provided in Supplementary Note 3.

**In situ irradiation experiments and TEM measurements**. The single-crystal Fe is provided by Metal Crystals and Oxides Ltd. The samples are cut into slices with a <100> normal plane by spark erosion. The standard 3 mm TEM discs are punched out and thinned mechanically to $50 \pm 100 \, \mu m$. After annealing in vacuum for an hour, the samples are slowly cooled down for further polishing into TEM thin foils, which are analyzed via TEM immediately for further experiments. The in situ ion irradiation and observation are carried out at the IVEM-Tandem Facility at Argonne National Laboratory. The irradiations are performed at 300 or 773 K with 100 or 150 keV $Fe^+$ ions to a maximum total dose in the range of $10^{18}$ ions $m^{-2}$ to $2 \times 10^{19}$ ions $m^{-2}$, respectively. The TEM observations are made at an operating voltage of 200 kV, using weak beam dark-field, kinematical bright-field, and dynamical two-beam diffraction conditions. The videos are made during the observation to record the whole kinetic process during and immediately after ion irradiation.

## Data availability
All data reported in this paper is available in the main text or the Supplementary Information.

## Code availability
The code or subroutines that support the findings of this study are available from the corresponding author, upon reasonable request. The parameters used for the present work are available in the Supplementary Information.

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

## Acknowledgements

We would like to acknowledge the financial support from the National MCF Energy R&D Program (Project No. 2018YFE0308101) and the National Natural Science Foundation of China (Project Nos. 12075141, 11675230, and 11427904). Computational resources for SAAMD simulations were provided by Institute of Modern Physics and ShanDong University.

## Author contributions

N.G. and F.G. designed the project and performed the simulations. Z.W.Y. designed the in situ irradiation experiments and performed the in situ TEM observations. N.G., Z.W.Y., H.Q.D., G.H.L., and F.G. performed the analysis. N.G. and F.G. wrote the paper. All authors discussed and commented on the manuscript.

## Competing interests

The authors declare no competing interests.
