## [Peer Review File · Nature Communications]

REVIEWER COMMENTS

Reviewer #1 (Remarks to the Author):

The submitted paper reports on atomistic simulations of dislocation loops in BCC Fe. Using the SAAMD method (developed by some of the co-authors) they find that a single loop in an otherwise perfect crystal under (presumably) full periodic boundary conditions, will undergo a series of habit-plane changes that together result in a net movement of the loop. The authors describe this as a new diffusion mechanism for such loops. The observed phenomenon is interesting. Some comments and questions are:

- 1) Generally very little information is given about both conventional and SAAMD simulations. For example, have the authors corrected for image interactions? There exist methods to do this. This could strongly effect the energetics and as a result the dynamics. Concerning table 2, which give the habit plane energies as a function of loop size, no information is given about the size of the simulation cell. Does it become larger for the larger loops? Irrespective of this, the numbers are possibly meaningless without image corrections.
- 2) There is no real explanation of the SAAMD method. Ref[14], in which it is described, is not well cited and therefore the method as is implemented in Ref.[14] appears not to be generally used. Moreover, such boost methods might fundamentally change the potential energy landscape resulting in incorrect processes being identified. It is therefore no surprise that somewhat similar results are achieved when the loop is stress driven.
- 3) Following on from this, how is fig. 4 obtained with SAAMD? No information is given about how the saddle points are obtained.
- 4) The comparison to experiment is quite superficial. For example the authors write "It is clear that the loop has an energy barrier between 0.9 to 1.2 eV ... which is comparable with recent experimental results [16]." In what way is a barrier energy comparable to experimental results? Ref. [16] is a private communication, which is not helpful. The claim of "Confirmation by in-situ irradiation and TEM measurements" is not justified. Firstly, the authors are well aware that the existence of 100 loops originates because of an elastic softening at high temperature. In other words, the energetics are quite different at high temperatures. In fact, in Fe, 100 loops generally do not occur at low temperature. Therefore making comparisons between low temperature model atomistic simulations (which cannot correctly simulate high temperatures) and high temperature experiment is difficult and possibly meaningless. This aspect is not at all mentioned in the paper. Secondly, the authors have not provided a convincing argument that the TEM observed dynamics is in fact a loop changing its habit plane. Thirdly, the experimental internal stress landscape is quite different from that of the model calculations, and therefore are quite likely to play a very strong role in loop mobility, and therefore a simple comparison is not justified. No mention of these aspects is made in the paper.
- 5) Whilst the results are interesting, the novelty of the findings are not clear. That a loop will jump between low-index habit planes is certainly not new. Also, that it is involved in the loops mobility has been studied before. See for example, the recent paper 10.1103/PhysRevMaterials.3.073805

Because of the above, I cannot recommend publication.

Reviewer #3 (Remarks to the Author):

Gao et al studied the diffusion mechanism of $\langle 100 \rangle$ interstitial dislocation loop in BCC Fe by combining self-adaptive accelerated molecular dynamics, classical molecular dynamics and in situ TEM. They revealed a new 1D diffusion mechanism for those dislocation loops that changes the habit plane between $\{100\}$ plane and $\{111\}$ plane.

Those findings are new and important. The manuscript reads well. In particular, their presented results from SSAMD, MD and in situ TEM provide well-connected and convincing evidence to support the conclusion in this manuscript. Before recommending for publication, I have two comments as below:

1) The atomistic simulation in this work used Ackland Fe potential. But the authors mentioned the other two potentials, Mendeleev (need to correct the name in Methods) and Dudarev-Derlet potential also work. So could the authors provide related simulation results using those potentials in Supporting Information? Especially the energy barrier for diffusion pathway (similar as Fig. 4). This can help to show the effect of potentials in diffusion mechanism.

2). As observed in in situ TEM, some $\langle 100 \rangle$ dislocation loop move, while some didn't. Please provide more explanation for such diffusion difference. More importantly, is there any statistical description for the mobility of those loops as learned from in situ TEM?

Responses to the reviewers

We thank the referees for their critical comments on our work and for providing valuable and constructive suggestions in the areas required clarification and improvement. We have addressed the comments and suggestions fully in the revision and our one-by-one responses to the referees' comments are given below.

Reviewer #1 (Remarks to the Author):

1) Generally very little information is given about both conventional and SAAMD simulations. For example, have the authors corrected for image interactions? There exist methods to do this. This could strongly effect the energetics and as a result the dynamics. Concerning table 2, which give the habit plane energies as a function of loop size, no information is given about the size of the simulation cell. Does it become larger for the larger loops? Irrespective of this, the numbers are possibly meaningless without image corrections.

Answer: We thank the referee for this comment. We definitely agree with the viewpoint of the referee about the possible image interaction and size effect on large loops. In fact, considering the length limitation of paper, although we have done the related examinations before the SAAMD simulations, we did not include enough information in the submitted version. In the revision, we clarified the related information by adding the following texts on pages 5 and 10:

“It should be noted that with increase in the number of the SIAs in a loop, the computational box is also increased accordingly, from the initial 20 to 50 unit cells (for a 171-SIA loop with the radius around 1.5 nm) along each direction to avoid the possible size effect.”

“The total number of atoms in this loop is around 69, which is 0.4% of the total number of atoms in the simulation box. Since the loop is located at the center of the box, the distance between the loop and its image is around 5.7 nm along the X direction and around 3.7 nm along the Y and Z directions because of the periodic boundary conditions. It should be noted that the strongest interaction between loops is along its Burgers vector direction, but decreases as a function of the reciprocal distance ($\sim 1/r^3$), much faster than an edge dislocation ($\sim 1/r$) [F. Kroupa. Journal de Physique Colloques, 27 (C3), pp.C3-154-C3-167 (1966)]. Considering the cutoff distance (around 0.54 nm) of the Fe potential used in this work, which is not a long-range correlation distance and is also much smaller than the half of box length, thus, the image interaction in this work can be neglected, as confirmed by the stress field of this loop shown in the supplementary materials. The formation energy of this loop calculated as a function of unit cells has also indicated that 17-18 unit cells along each direction are large enough to avoid the image interaction, as shown in the

supplementary materials.”

2) There is no real explanation of the SAAMD method. Ref. [14], in which it is described, is not well cited and therefore the method as is implemented in Ref.[14] appears not to be generally used. Moreover, such boost methods might fundamentally change the potential energy landscape resulting in incorrect processes being identified. It is therefore no surprise that somewhat similar results are achieved when the loop is stress driven.

Answer: We thank the referee for this comment. Following your suggestion, we provide more information about the SAAMD method used in this work and also revise the text to include how to use SAAMD for the present simulations on page 10. This method is developed by focusing on the rugged energy surface with a high energy barrier, although it can also be used for determining an energy on a smooth surface, such as free surface diffusion, as we have demonstrated in Ref. [15] in the revised manuscript. The basic idea for all accelerated molecular dynamics is to fill the potential in order to increase the probability for a rare event to jump and overcome the energy barrier, as explained by A. Voter who firstly developed the hyperdynamics to accelerate the classical molecular dynamics method. SAAMD is developed by keeping the same idea. Therefore, the change on the potential energy landscape induced by accelerated molecular dynamics method would not affect the underlying physics process but only increase the probability for state to jump over the energy barrier. The related information has been stated in Ref. [15]. The migration of helium-vacancy clusters in materials using the SAAMD method has also been confirmed by LANL researchers [B. P. Uberuaga, E. Martínez, D. Perez, A. F. Voter, Computational Materials Science, 147, 282 (2018)].

“The SAAMD was developed to model the infrequent atomic scale events, especially those events that occur on a rugged free-energy surface, for example, the diffusion of complexed radiation defects in materials. The boost potential is slowly increased to accelerate the dynamics by slowly filling the potential as suggested by Voter [26]. However, the key of this filling potential to slowly boost dynamics does not to alter the energy barrier, thus not affecting the underlying physics, but just increasing the probability for the event to occur by overcoming the high energy barrier. This approach is self-evolving, where the total displacement of the system at a given temperature is used to construct a boost-potential. The method can be used to study not only the migration of atoms on the free surface, but also the migration of helium-vacancy cluster in materials, which has been confirmed by LANL researchers [27]. Thus, considering the possible high energy barrier for the diffusion of a $\langle 100 \rangle$ loop, the SAAMD is applied for this work.”

“To use the SAAMD method, the active volume (AV) should be selected first. In the present work, considering the possible migration of a $\langle 100 \rangle$ loop along its Burger vector, a cylindrical AV is selected around the $\langle 100 \rangle$ loop with a radius 20% larger

than that of the loop for the SAAMD simulation, within which all the atoms are accelerated, as shown in Fig. 7. Following the Ref. [15], the criteria to determine the state change of the $\langle 100 \rangle$ loop is introduced by controlling the maximum displacement of an atom (D_{\max}) in the AV. The maximum displacement is set to 2.0 Å (~ 80% of Burgers vector), determined by multiple pre-simulations, for the present study. Once the maximum displacement is reached, the boost potential is removed and classical MD is then used for another 20 ps with a time-step of 1 fs, after which, MS is applied for the system to reach a new local stable configuration. Then, the SAAMD is applied again to accelerate the atoms in the AV. The temperature for the SAAMD simulations is 300 K.”

3) Following on from this, how is fig. 4 obtained with SAAMD? No information is given about how the saddle points are obtained.

Answer: We thank the referee for this question. Fig.4 is calculated using the nudge elastic band (NEB) method, which is a general method to calculate the energy barrier with a jump. According to this method, the key steps to obtain Fig.4 is to determine the initial and final states for one jump. For example, in Fig.4, the initial state is the loop located on a habit plane of the $\{100\}$ and the final state is on a habit plane of the $\{130\}$ at the 1st jump. The second jump is from the initial $\{130\}$ habit plane to the final $\{120\}$ habit plane, followed which the third jump occurs to the final $\{110\}$ habit plane. However, it should be emphasized that these initial and final states, as the local minimal energy states, have to be obtained by SAAMD simulations. Thus, the energy barriers between these initial and final states can be calculated by the NEB method. Following your suggestion, we also provide the saddle point states explored by the SAAMD and used by the NEB method in the revised supplementary materials.

“In Fig.4, there are saddle point states between the $\{100\}$, $\{130\}$, $\{120\}$ and $\{110\}$ habit planes. The configurations of these states are shown in Fig.S3. It is clear that these states are also possible metastable habit planes, which may be not expressed exactly by the integers $\{hk0\}$, but are still located between different habit planes. However, the saddle point states are explored by the SAAMD and used by the NEB method to determine the energy barriers.”

Fig.S3: Configurations of saddle point states during the rotation of the habit planes from the $\{100\}$ to the $\{110\}$. For clarity, the habit planes of the $\{100\}$, $\{130\}$, $\{120\}$

and $\{110\}$ are represented by the red dash lines.

4) The comparison to experiment is quite superficial. For example the authors write "It is clear that the loop has an energy barrier between 0.9 to 1.2 eV which is comparable with recent experimental results [16]." In what way is a barrier energy comparable to experimental results? Ref. [16] is a private communication, which is not helpful. The claim of "Confirmation by in-situ irradiation and TEM measurements" is not justified.

Answer: We thank the referee for this question. The results from Prof. Arakawa group were obtained by *in-situ* TEM observation. These results have not been published. Following your suggestion and also to avoid the uncertain conclusion, we deleted above claim in the revised version. However, we believe that their results will be published eventually.

5) Firstly, the authors are well aware that the existence of 100 loops originates because of an elastic softening at high temperature. In other words, the energetics are quite different at high temperatures. In fact, in Fe, 100 loops generally do not occur at low temperature. Therefore making comparisons between low temperature model atomistic simulations (which cannot correctly simulate high temperatures) and high temperature experiment is difficult and possibly meaningless. This aspect is not at all mentioned in the paper.

Answer: We thank the referee for this comment. We agree with the referee that the elastic softening of iron at high temperature may be one of the possible mechanisms of $\langle 100 \rangle$ loop formation. However, there exists different mechanisms for its formation. For example, through the high energy cascade or the loop interaction of two $1/2\langle 111 \rangle$ loops, a $\langle 100 \rangle$ loop can also be formed even at lower temperatures. The recent experimental results also confirmed that at temperature around 270 K, the $\langle 100 \rangle$ loops have been formed, although the $1/2\langle 111 \rangle$ loops dominate in terms of the loop density. At around 310 K, the diffusion of $\langle 100 \rangle$ loop has been concluded from the TEM observation [14]. If only from the energy viewpoint, we agree with the referee that the energy state of a $\langle 100 \rangle$ loop at low temperatures is different from that at high temperatures. However, considering the high binding energy of a SIA to the $\langle 100 \rangle$ loop, it has very low probability to observe the decomposition of the loop at the temperatures from low to high. Thus, because of its stability at different temperatures, it is not difficult just to compare the kinetics from the atomistic simulation results with the experimental results. To better address the referee's comment, we have also performed the SAAMD to simulate a case at a higher temperature around 600 K, which indicates the same habit plane rotation process occurred during its 1D diffusion. The comparison of each atom migration behavior in a loop with the experimental observation is very difficult; however, it is useful to explore the main feature of a $\langle 100 \rangle$ loop during its migration through the comparison

between the simulations and in-situ TEM observations, which is also one of the main purposes in the present work. Following your suggestion, we emphasize in the revised version that the comparison between the atomistic simulations and the in-situ TEM observations would not focus on the kinetic of each atom, but on the main feature of the habit plane changes during the migration of a <100> loop. The related explanation is provided on page 8 in the revised manuscript.

“It should be noted that although it is difficult to compare the kinetic process of each atom within the loop, it is still useful and helpful to compare the atomistic simulation results with the *in-situ* TEM observations at 773 K, which can be used to explore the main features of the loop, i.e. the rotation of its habit planes driving its 1D diffusion.”

“Also, the SAAMD simulations at 600 K present the same results to those observed at the lower temperature; the rotation of the habit planes of a <100> loop during its 1D diffusion process. From the recent experimental results [14], the diffusion of a <100> loop observed by the TEM observations can be then used to compare with the present simulation results.”

6) Secondly, the authors have not provided a convincing argument that the TEM observed dynamics is in fact a loop changing its habit plane. Thirdly, the experimental internal stress landscape is quite different from that of the model calculations, and therefore are quite likely to play a very strong role in loop mobility, and therefore a simple comparison is not justified. No mention of these aspects is made in the paper.

Answer: We thank the referee for these comments. Let us try to address these questions as following.

The dislocation contrast is determined by $\mathbf{g} \cdot \mathbf{R}$, for pure edge dislocation;

$$\mathbf{R} = \frac{1}{2\pi} \left(\mathbf{b}\phi + \frac{1}{4(1-\nu)} \{ \mathbf{b}_e + \mathbf{b} \times \mathbf{u} (2(1-2\nu)\ln r + \cos 2\phi) \} \right)$$

The $\mathbf{g} \cdot (\mathbf{b} \times \mathbf{u}) \neq 0$ may give a residual contrast of the dislocation though $\mathbf{g} \cdot \mathbf{b} = 0$, and so only the dislocation line direction changes cause the contrast changes, which are exactly indicated in the figure 6. However, the size of a loop (loop contour) won't change due to the difficulty of climb. From fig.6 a -f, the segments of the loop 1 appear and disappear alternatively would only result from the change of their habit plane. We are not able to directly indicate to which habit plane they change from the (001) pane, but the simulation should provide the clue. That is why it is useful to combine the computer simulation with experiment to explore a complicated mechanism sometimes.

Regarding the internal stress, the experimental internal stress landscape may be different from that of the model calculations. The dependence of loop mobility on possible internal stress are analyzed below.

Forces on Dislocations: Peach-Koehler eq., σ_{ij} can be image stress from free surface or the elastic field from another dislocation.

$$f = (b \cdot \sigma_{ij}) \times \xi$$

$$\begin{bmatrix} f_1 \\ f_2 \\ f_3 \end{bmatrix} = |b_1 \ b_2 \ b_3| \begin{bmatrix} \sigma_{11} & \sigma_{12} & \sigma_{13} \\ \sigma_{21} & \sigma_{22} & \sigma_{23} \\ \sigma_{31} & \sigma_{32} & \sigma_{33} \end{bmatrix} \times \begin{bmatrix} \xi_1 \\ \xi_2 \\ \xi_3 \end{bmatrix}$$

Two dislocations may interact via their own elastic field.

The f_x is the attraction force between two opposite segments from different $\langle 100 \rangle$ dislocations, which will drive them to move towards each other. Basically, we can estimate this (internal) stress through their separation distance. Therefore, some $\langle 100 \rangle$ dislocation loops move, while some would not move, depending on the stress applied on each loop.

However, it should be noted that we are not focusing on which way the loop is triggered to migrate along its Burgers vector, but instead on the possible diffusion mechanisms of a $\langle 100 \rangle$ loop, which has not been explored previously. For example, for a vacancy diffusion, the experimental internal stress landscape is different from that of the simulation model calculations, but the diffusion mechanism is similar, i.e. the vacancy jumps to one of the nearest atoms. Furthermore, both experiments and simulation model demonstrated that a $\frac{1}{2}\langle 111 \rangle$ dislocation loop in Fe migrates one-dimensionally even though the stress landscape is different (K. Arakawa, K. Ono, M. Isshiki, K. Mimura, M. Uchikoshi, H. Mori, Science, 318 956-959 (2007)). Similarly, we tried to explore the new diffusion mechanisms of a $\langle 100 \rangle$ loop, which can be obtained both from the in-situ observations and SAAMD simulations. Following your suggestion, we provide a further explanation about this comparison on page 8 in the revised manuscript.

“The video clearly shows the rotation of the loop from its habit plane the $\{100\}$ to the other habit planes. Combining with the simulation results and crystal geometry, the process from $\{130\}$, to the $\{120\}$, and finally to the $\{110\}$, is possible.”

“Although the internal stress landscape from the atomic simulations may be different from that in the experimental sample, the results from both methods indicate the rotation of the habit plane of a $\langle 100 \rangle$ dislocation is a one of the major mechanisms

for its diffusion. Thus, the present work is not to focus on which way the loop is triggered to diffuse along its Burgers vector, but instead on the possible diffusion mechanisms of the loop, which has not been explored previously.”

7) Whilst the results are interesting, the novelty of the findings are not clear. That a loop will jump between low-index habit planes is certainly not new. Also, that it is involved in the loops mobility has been studied before. See for example, the recent paper 10.1103/PhysRevMaterials.3.073805.

Answer: Thank you referee, but we respectively disagree with this comment. In PRM paper, the diffusion was focusing on the $1/2\langle 111 \rangle$ loops, whose diffusion process has been well known for long time, i.e. the rotation of its habit plane along the same rotation axis. It should be noted that the diffusion of a $\langle 100 \rangle$ interstitial dislocation loop has been recently explored by experiments, as did in this work and in Ref. [13]. To our knowledge, it is the first time to investigate the diffusion mechanism of a $\langle 100 \rangle$ dislocation loop in the current work, which is different from the results of the paper recommended by the referee. The rotation of the habit plane of a $\langle 100 \rangle$ loop is not along the same rotation axis, which is different from that of a $1/2\langle 111 \rangle$ loop. The different mechanisms for the $1/2\langle 111 \rangle$ and $\langle 100 \rangle$ dislocation loops to diffusion represent the novelty of the current study. Furthermore, the possible impacts of the $\langle 100 \rangle$ loop diffusion on radiation damage have also been proposed based on the present simulation results. We totally agree with the referee that we need to better clarify this point in the revision. The novelty of the current study, along with the difference between diffusion mechanisms of a $\langle 100 \rangle$ loop and a $1/2\langle 111 \rangle$ loop, have been clearly expressed in the revised paper on pages 6 and 9. We also included above paper in the introduction section above the properties of a $1/2\langle 111 \rangle$ loop.

“The diffusion mechanism of the $\langle 100 \rangle$ loop is different from the $1/2\langle 111 \rangle$ loop, particularly relating to the intersection of these habit planes. As the habit plane of the $1/2\langle 111 \rangle$ loop intersects at a line, whereas the $\langle 100 \rangle$ loop intersects at a point. Thus, the degree of freedom of these two loops during their diffusion is different. The related atomic arrangement during the diffusion is also different, which results in the different stress or strain field during the diffusion. The possible effects of these differences on the radiation damages have been discussed in the following section.”

“However, based on the current study, if the diffusion property of a $\langle 100 \rangle$ loop is taken into account, especially the intersection of its habit planes as a point during its diffusion, some important effects of $\langle 100 \rangle$ loops on radiation damage need to be reconsidered: (1) high efficiency for $\langle 100 \rangle$ loops to absorb the point defects and small defect clusters due to the stress field variation during their diffusion, resulting in the quick loop growth (as indicated by Fig.6 in Ref. [14]), and the increase in the pinning effect and, thus radiation hardening; (2) ...”

Reviewer #3 (Remarks to the Author):

Gao et al studied the diffusion mechanism of $\langle 100 \rangle$ interstitial dislocation loop in BCC Fe by combining self-adaptive accelerated molecular dynamics, classical molecular dynamics and in situ TEM. They revealed a new 1D diffusion mechanism for those dislocation loops that changes the habit plane between $\{100\}$ plane and $\{111\}$ plane.

Those findings are new and important. The manuscript reads well. In particular, their presented results from SSAMD, MD and in situ TEM provide well-connected and convincing evidence to support the conclusion in this manuscript. Before recommending for publication, I have two comments as below:

Answer: We thank the referee for providing suggestions to improve our manuscript and for recommending its publication.

1) The atomistic simulation in this work used Ackland Fe potential. But the authors mentioned the other two potentials, Mendelev (need to correct the name in Methods) and Dudarev-Derlet potential also work. So could the authors provide related simulation results using those potentials in Supporting Information? Especially the energy barrier for diffusion pathway (similar as Fig. 4). This can help to show the effect of potentials in diffusion mechanism.

Answer: We thank the referee for this comment. Following your suggestion, we provide these calculations based on different potentials with NEB method. The saddle point states are also provided in the supporting information.

“From Fig.S4, although the exact energy barrier is different from different potentials, the general trend during the rotation is similar from these three potentials.”

Fig.S4: Energy barrier calculated with Ackland04, Mendeleev and Dudarev-Derlet Fe potentials for a <100> loop with a diameter of around 1.6 nm.

2). As observed in in situ TEM, some <100> dislocation loop move, while some didn't. Please provide more explanation for such diffusion difference. More importantly, is there any statistical description for the mobility of those loops as learned from in situ TEM?

Answer: Thank the referee for above comment and question. We agree with the referee that the loops move or not, which depends on the applied stress from other sources, for example, another dislocation.

Forces on Dislocations: Peach-Koehler eq., σ_{ij} could be the image stress from free surface or the elastic field from another dislocation.

$$f = (b \cdot \sigma_{ij}) \times \xi$$

$$\begin{pmatrix} f_1 \\ f_2 \\ f_3 \end{pmatrix} = \begin{vmatrix} b_1 & b_2 & b_3 \end{vmatrix} \begin{vmatrix} \sigma_{11} & \sigma_{12} & \sigma_{13} \\ \sigma_{21} & \sigma_{22} & \sigma_{23} \\ \sigma_{31} & \sigma_{32} & \sigma_{33} \end{vmatrix} \times \begin{pmatrix} \xi_1 \\ \xi_2 \\ \xi_3 \end{pmatrix}$$

Two dislocations may interact via their own elastic field.

The f_x is the attraction force between two opposite segments from different <100> dislocations, which will drive them to move towards each other. Basically, we can estimate this (internal) stress through their separation distance. Therefore, some <100> dislocation loops move, while some would not move, depending on the stress applied on each loop.

Based on the present in-situ TEM measurements, although it is still challenge to obtain the statistical description for the different radiation defects, which lead to the <100> loop to move from in situ TEM, we could conclude once the stress from other radiation defects is higher than its critical stress, the loop would rotate between different habit planes, resulting in the 1D diffusion. Following your suggestion, we included the related information in the revised manuscript on page 8 to provide one possible explanation for immobile loops.

“In in-situ experiments, the motion of a <100> dislocation loop is driven by the Peach-Koehler force generated by other sources, such as surrounding dislocations and irradiated defects. However, if this force is not large enough, the rotation of their habit

planes does not occur, and thus, some dislocation loops remain immobile.”

Reviewers' comments:

Reviewer #1 (Remarks to the Author):

The submitted manuscript has somewhat improved. In relation to my specific comments:

1) I do not agree with their conclusion. Image corrections can be very important, affecting the energetics of the defect - especially as a function of orientation with respect to the simulation cell. A $1/r^3$ dipolar interaction is long range in three dimensions with an energy that will depend on system size. Because of defect anisotropy, convergence is nevertheless possible - but can be slow with respect to increasing system size. Finally this has nothing to do with the finite range of the potential! This is an anisotropic elastic effect.

2) My question has not been answered. To say "Therefore, the change on the potential energy landscape induced by accelerated molecular dynamics method would not affect the underlying physics process but only increase the probability for state to jump over the energy barrier." is unjustified. It is true, that the overall barrier energy is weakly affected by such methods but the all important prefactor can strongly change when the landscape is strongly modified as is the case for the SAAMD method. The correct acceleration of collective changes in structure is a complex and unsolved problem. To be fair, all methods suffer from this uncertainty.

3) OK

4) OK

5) I don't understand the authors response. The reality is that due to elastic softening, the empirical potentials used in such simulations (and the associated observations and conclusions) cannot be applied to high temperature since they incorrectly describe the elastic field of a defect. The new simulations at 600K are therefore not useful.

6) The basic theory stated is correct, but as the authors admit the comparison to experiment is weak - their simulations motivate their interpretation. It is my view that the added statements (marked in red) are not justified, the internal landscape of an irradiated structure is vastly different situation as that of an isolated defect in an otherwise perfect crystal. To conclude this point, I do think there is great worth to understand defects in simplified geometries, but to compare to experiment in a quantitatively meaningful way is very difficult to do. My view is that such a comparison is not achieved in the present work. The second referee points this out quite nicely, since he/she asks why do most loops not change their habit plane to which the authors' respond that in these cases, it is due to internal stress! Anything can be said. Two detailed comments are a) the truth of the statement "However, the size of a loop (loop contour) won't change due to the difficulty of climb." is not obvious to me, given that this is an irradiated material full of point defects, and b) the text "Thus, the present work is not to focus on which way the loop is triggered to diffuse along its Burgers vector, but instead on the possible diffusion mechanisms of the loop, which has not been explored previously" is not understandable to me.

7) OK

Given the above, unfortunately I cannot recommend publication in a journal such as Nature Communications.

Reviewer #2 (Remarks to the Author):

The reply letter has solved my comments/questions. So I recommend it for publication.

Rebuttal Letter

Our one-by one responses to the referees' comments are provided below. The current correction in the revised version is highlighted by the purple color (the previous correction is highlighted by the red color).

The second referee suggested to publish the paper in Nature Communications without further comments.

The first referee:

1) I do not agree with their conclusion. Image corrections can be very important, affecting the energetics of the defect - especially as a function of orientation with respect to the simulation cell. A $1/r^3$ dipolar interaction is long range in three dimensions with an energy that will depend on system size. Because of defect anisotropy, convergence is nevertheless possible - but can be slow with respect to increasing system size. Finally this has nothing to do with the finite range of the potential! This is an anisotropic elastic effect.

Answer: We thank the referee for this question. We agree with the referee that the image correction is very important for the molecular dynamics simulations of dislocations and dislocation loops. However, there is a well-established theory, as described in the book of "Comprehensive Nuclear Materials, volume 1, pp 249-265 Amsterdam: Elsevier" by Profs. Wei Cai, Ju Li and Sidney Yip. In one of their simulations, an edge dislocation was introduced into a box with two free surfaces to avoid the dislocation dipole effect on the simulation results. What they found is that as the dislocation moves to an equivalent lattice site along its Burgers vector direction, the image force on the dislocation from the boundary condition can be neglected if the simulation box is large enough and the formation energy does not change. We have used the same method to evaluate the image corrections, and the results suggest that our simulation block is large enough to eliminate the image effect, as shown in Fig.S3 in revised supplementary materials (also see the figure below).

Figure S3: Formation energy of a dislocation loop located at different positions along its Burgers vector, where the unit of distance is given by its Burgers vector.

In addition, we provided the detailed results about the size effect on the formation energy of a dislocation loop to clearly show that the image correction can be neglected with the present box size in our previous response (also see the details in Fig.S2 in the supplementary materials).

Fig.S2: Formation energy of a $\langle 100 \rangle$ loop with a radius of 1.0 nm as a function of the unit cells of computational box.

In fact, the previous MD simulations of dislocations and dislocation loops, for example, S. J. Zhou et.al. *Science*, 279 1525(1998), J. P. Chang et.al. *Materials Science and Engineering A*, 309 160(2001), J. Li et.al., *Nature* 418 307(2002), J. Marian, W. Cai, V. V. Bulatov, *Nature Materials*, 3, 158 (2004), H. Van Swygenhove, P. M. Derlet, A. G. Froseth, *Acta Materialia*, 54 1975 (2006), Q. Peng et.al., *Nature Communication*, 9 1(2018), have presented important results for dislocations or dislocation loops through MD simulations. These results show that the image corrections can be neglected when the system size is large enough.

2) My question has not been answered. To say "Therefore, the change on the potential energy landscape induced by accelerated molecular dynamics method would not affect the underlying physics process but only increase the probability for state to jump over the energy barrier." is unjustified. It is true, that the overall barrier energy is weakly affected by such methods but the all important prefactor can strongly change when the landscape is strongly modified as is the case for the SAAMD method. The correct acceleration of collective changes in structure is a complex and unsolved problem. To be fair, all methods suffer from this uncertainty.

Answer: We thank the referee for this comment. We have described the method used in our paper in details. The SAAMD method is different from other accelerated MD approaches. In the SSAMD, the boost potential is slowly increased to accelerate the dynamics by slowly filling the potential, where the total displacement of the system at a given temperature is used to construct a boost-potential. The migration mechanism of helium/vacancy clusters explored by the SAAMD was confirmed by LANL researchers who used so called parallel replica dynamics (ParRep) (please note that this method accelerates dynamics without boosting the potential). Their study well demonstrated that the SAAMD provides the same dynamics as ParRep without boosting potential. Because the boost potential is slowly increased to accelerate the dynamics, the SAAMD should not have a significant effect on pre-factor and energy barrier for jump. However, following the referee's suggestion, we deleted "thus not affecting the underlying physics" in the revised manuscript to avoid the possible misleading about the effect of AMD on pre-factor of event jumps on energy landscape.

5) I don't understand the authors response. The reality is that due to elastic softening, the empirical potentials used in such simulations (and the associated observations and conclusions) cannot be applied to high temperature since they incorrectly describe the elastic field of a defect. The new simulations at 600K are therefore not useful.

Answer: Thanks for this comment. We have to make a clear statement again that the current study did not focus on how the $\langle 100 \rangle$ loops are formed in bcc Fe, but on the migration mechanism of the $\langle 100 \rangle$ loops.

In terms of elastic softening, we have tested our simulations with the Dudarev-Derlet Fe potential, which considers the elastic softening properties of Fe with increasing temperature. However, the migration mechanism for the $\langle 100 \rangle$ dislocation loops is very similar, with a slightly different energy barrier (Fig.S5 in the revised supplementary materials), which suggests that the elastic softening is not critical for our current simulations. Previously, Dudarev, Bullough, and Derlet used the concept of elastic softening to explain the stability of dislocation loops in bcc Fe. According to this theory, the $\langle 100 \rangle$ loops could be "unconditionally stable" with temperatures higher than 550 K in iron and increasingly favorable at the temperatures higher than 300 K, (Note that Fig. 4 in their paper showed that this temperature should

be 350 °C). However, the recent experimental results confirmed that at a temperature around 270 °C, the <100> loops have been observed to form in BCC iron. At the temperature around 310 °C, the diffusion of <100> loops has been concluded from the TEM observations (J. Chen et al., Journal of Nuclear Materials 503, 81 (2018)). Thus, the elastic softening theory cannot be used to explain these experimental results. In fact, the formation of <100> loops at lower temperatures has also been reported experimentally in other BCC materials. For example, the irradiation with H⁺ ions at room temperature induced the formation of around 8% <100> loops in BCC vanadium [L.J. Cui, et.al, Acta Physica Sinica, 65,066102(2016)]. BCC molybdenum irradiated by Sb⁺, Sb²⁺ and Sb³⁺ at room temperature also clearly show the formation of <100> loops [C. A. English and J. L. Jenkins, Philosophical Magazine, 90, 7-14 (2010)].

In addition, we simulated the migration of <100> dislocation loops, but not their formation. The simulations performed at 600 K clearly indicate the same migration mechanism of a <100> loop, showing the temperature would not affect the migration mechanism. In fact, the empirical Fe potential used in this work has been also applied by other studies with temperature up to 600 K. How the elastic softening affects the motion of <100> dislocation loops is absolutely unclear, which is however not the topic of this paper.

6) The basic theory stated is correct, but as the authors admit the comparison to experiment is weak - their simulations motivate their interpretation. It is my view that the added statements (marked in red) are not justified, the internal landscape of an irradiated structure is vastly different situation as that of an isolated defect in an otherwise perfect crystal. To conclude this point, I do think there is great worth to understand defects in simplified geometries, but to compare to experiment in a quantitatively meaningful way is very difficult to do. My view is that such a comparison is not achieved in the present work. The second referee points this out quite nicely, since he/she asks why do most loops not change their habit plane to which the authors' respond that in these cases, it is due to internal stress! Anything can be said. Two detailed comments are a) the truth of the statement "However, the size of a loop (loop contour) won't change due to the difficulty of climb." is not obvious to me, given that this is an irradiated material full of point defects, and b) the text "Thus, the present work is not to focus on which way the loop is triggered to diffuse along its Burgers vector, but instead on the possible diffusion mechanisms of the loop, which has not been explored previously" is not understandable to me.

Answer: Thanks referee for this comment. It is true that the internal landscape of an irradiated structure is different from that of an isolated defect in a perfect crystal. However, according to the present experimental observation details and unique explanation in terms of the dislocation $\mathbf{g} \cdot \mathbf{R}$ contrast criterion for a pure edge dislocation, where from figs.6 a-f, the segments of the loop 1 that appear and disappear alternatively would be only resulted from the change of their habit planes, according to the well-known contrast theory in TEM measurements, which presents

the direct link between computer simulation and experimental TEM observation. That is why the computer simulation was demanded to explore the complication of its migration mechanism. Although it is difficult to get the whole picture of the internal landscape of irradiated materials, the unique link between experiments and computer simulations determines the possible diffusion mechanism of a $\langle 100 \rangle$ loop through these two methods.

For two detailed comments:

a) the truth of the statement "However, the size of a loop (loop contour) won't change due to the difficulty of climb." is not obvious to me, given that this is an irradiated material full of point defects.

This statement is related with the present experimental condition, instead of explaining all experimental results. Furthermore, even in an irradiated material full of point defects, given that at low temperatures (e.g. several K), the size of loop would be difficult to be changed because the movement of point defects to a loop is prohibited. Therefore, it would be reasonable to understand this statement according to our experimental details.

b) the text "Thus, the present work is not to focus on which way the loop is triggered to diffuse along its Burgers vector, but instead on the possible diffusion mechanisms of the loop, which has not been explored previously" is not understandable to me.

We tried to make a clear statement that the topic of this paper is to explore the diffusion mechanism of a $\langle 100 \rangle$ loop. We did not focus on the whether the diffusion of loop is induced by stress or temperatures.

REVIEWER COMMENTS

Reviewer #1 (Remarks to the Author):

The submitted manuscript has somewhat improved. Concerning:

1) I am confused with fig. S3 and the simulation. Also, for which system size is this. I would have expected a small and scattered gradient, not a gradient that appears identically zero, may be it is choice of the vertical scale. Also, the discussed method involved surfaces, I was under the assumption that these simulations were performed under periodic boundary conditions. Fig. S2 appears reasonable. A more convincing plot would have been a plot of the energetics as a function of system size for different habit planes. If there are surfaces (as implied by fig. S3), I am surprised that the energies change so little.

One may always study dislocation processes in finite systems (as is done in the publication list provided by the authors) however when one is interested in small energy differences, such as in the present work, image effects must be discussed.

2) If the SAMMD method has been validated using parallel replica methodologies, this is quite an important result for the author's method and something readers should know. The work has to be cited, whether published or not.

5) To my knowledge, there exists no empirical potential that correctly describes the high temperature elastic properties of bcc Fe. Having correct elastic properties is vital to obtaining the correct energetics of all defects at this high temperature. This should be clearly stated in the manuscript.

6) OK concerning 6a. The answer to 6b remains confusing.

To conclude, 1) the surface issue needs to be clarified and 2) the appropriate citation needs to be included. 5) Cannot be further resolved until potentials that are able to describe the elastic phenomenon become available. Until then, all high temperature MD work must be qualified with the fact that the elastic constants are poorly described.

Reviewer #3 (Remarks to the Author):

Contrary to previous assumptions, the manuscript by Gao et al predicts that a type of interstitial loop in BCC Fe is in fact mobile at room temperature. If true, this finding would represent a significant advance in our understanding of irradiated materials. However, the simulations are not entirely convincing. More detailed comments are provided below.

1). First, I am satisfied that the system size used in this study is sufficiently large such that effects of neighboring defects from the periodic boundary conditions do not substantially affect the results. (However, I do agree that the cutoff of the potential is irrelevant in this discussion and that sentence in the paper – line 306 – should be removed.) Second, it is true that some EAM potentials do not capture the energetics of interstitials well, but the authors have tested several different potentials and apparently there are no obvious differences in the results.

2). The main concern I have – and it is a major one – was discussed (I think) in comment number 6 of the previous reviewers. The simulations show that the $\langle 100 \rangle$ loops can adopt different habit planes from thermal fluctuations. That would mean, however, that different habit planes should be observed in experiment. I admit I am not completely up to date on experimental findings, but it seems to me only the $\langle 100 \rangle$ habit has been found. If so, then this discrepancy is a serious problem and calls into question the entire simulation methodology.

The boost potential scheme originally proposed by Voter works well for cases such as surface diffusion, where every adatom site on the surface has the same potential energy landscape. In these cases it is straightforward to generate a boost that achieves acceleration, but does not alter the potential energy in the vicinity of the saddle point. In recent years MD techniques have been developed where the boost can be used for varying energy landscapes. The SSAMD is one such technique. In the original SSAMD paper the authors examined the case of He atom clustering. Here you would expect the energy landscape of He to vary depending on the presence of nearby He atoms, but the energy landscape, I would think, does not vary significantly. In the case of a PIDL I would expect the energy landscape to change substantially for atoms all along the edge of the loop. Therefore, how do we know the SSAMD is not introducing errors? Also, as an interstitial atom jumps out of the plane of the loop, then, again, I would assume the potential energy surface for that atom has now changed considerably. Is this accounted for in the SSAMD? Should it be?

In the SSAMD two parameters are used to compute the (time varying) boost potential. One is a prefactor E_b and the second is a cutoff distance q such that the boost potential goes to zero when the cutoff is reached. The authors never state what E_b is or how it is obtained. They should. More important, the authors have used a maximum cutoff of $q=2A$. This value seems completely incorrect. The nearest neighbor interstitial distance in BCC iron is considerably less than $2A$ and therefore during an interstitial hop a bias potential is applied during the entire path?? Since the saddle point will be roughly half way along the path the value of q seems to be too large by at least a factor of two. I realize there is a word limit on Nature Comm papers, but this point needs to be explained in more detail. How did the authors choose E_b and max q ? What simulations were used to set these parameters? Can the same parameters be used for the PIDL case? How do we know? Perhaps the authors can re-run the simulation leading to movie 1, but vary the SSAMD parameters over a large range and verify that the changing habit plane is not an artifact of the simulation method.

3). I do not understand the discussion surrounding eq. 1. The authors state the MD simulation was modified using this equation. How exactly is the equation used? How is a jump frequency incorporated into a MD simulation? What is the nature of the stress applied? Is it a hydrostatic pressure? Uniaxial tension? What is the energy barrier in this equation? Is it the energy barrier between the PIDL on different habit planes? Finally what is the prefactor?

4). On line 150 the authors state " . . . the $\frac{1}{2}\langle 111 \rangle$ loop intersects at a line, whereas the $\langle \text{loop} \rangle$ intersects at a point." What does "intersect" mean here? What is intersecting with what?

REVIEWER COMMENTS

Reviewer #1 (Remarks to the Author):

The submitted manuscript has somewhat improved. Concerning:

1) *I am confused with fig. S3 and the simulation. Also, for which system size is this. I would have expected a small and scattered gradient, not a gradient that appears identically zero, may be it is choice of the vertical scale. Also, the discussed method involved surfaces, I was under the assumption that these simulations were performed under periodic boundary conditions. Fig. S2 appears reasonable. A more convincing plot would have been a plot of the energetics as a function of system size for different habit planes. If there are surfaces (as implied by fig. S3), I am surprised that the energies change so little.*

One may always study dislocation processes in finite systems (as is done in the publication list provided by the authors) however when one is interested in small energy differences, such as in the present work, image effects must be discussed.

Answer: Thank referee for these comments. In Fig. S3, the fluctuation is very small with the maximum energy difference less than 1.0×10^{-5} when the loop is located at the different positions. Because of the large vertical scale and results taken with **3 digits after the decimal point**, it shows there is no gradient appeared in Fig. S3. Following your suggestion, we also further calculated the energies for a loop with different habit planes, as shown in the updated Fig. S3 in the supplementary materials.

Following your suggestion, we have included the discussion about the image effect in the revised paper by using the equations developed by Cai et. al. for a prismatic dislocation loop in a box under the periodic boundary condition. According to these equations, we can estimate the ratio of image force contribution to the total self-energy of the dislocation loop studied in this work, which is around 3.6×10^{-6} , similar to the results of a dislocation loop in a HCP lattice studied by H. J. Hu (H. J. Chu, E. Pan, X. Han, J. Wang and I. J. Beyerlein, Journal of the Mechanics and Physics of Solids, 60, 418, 2012).

We have clearly indicated this in both the supplementary materials (on pages 4 and 5) and the revised manuscript (on page 11), thus addressing the referee's concern.

2) *If the SAMMD method has been validated using parallel replica methodologies, this is quite an important result for the author's method and something readers should know. The work has to be cited, whether published or not.*

Answer: Thank you for the suggestion. Yes, the worked mentioned has been published. We have cited the work by Dr. Uberuaga from LANL as Ref [34] in our paper on page 12, which presented the same results for a helium-vacancy cluster to migrate in tungsten as studied by our SAAMD method.

5) *To my knowledge, there exists no empirical potential the correctly describes the high temperature elastic properties of bcc Fe. Having correct elastic properties is vital to obtaining the correct energetics of all defects at this high temperature. This should be clearly stated in the manuscript.*

Answer: Thank you for the suggestion. We have stated this clearly in the revised manuscript on pages 8 and 9.

“The reason for above difference for internal stress may be from: (1) uncertainty of energetics of all defects at higher temperatures induced by the empirical potential; until now there is no one empirical

potential that can be applied for exactly describing the high temperature elastic properties of bcc Fe, which is vital to obtaining the correct energy landscape of defects at high temperature; (2) the sample surface effect, applying additional stress field on the loop, may be different from the state obtained with periodic boundary condition used in the simulations.”

6) *OK concerning 6a. The answer to 6b remains confusing.*

Answer: Thank you again for your suggestion. In previous answers, we have clarified the internal stress that may be different from that of the model calculations, and provided the corresponding explanation. On pages 8 and 9 in the revised manuscript, we included more discussions and possible effects both from the empirical potential, which could not correctly describe the elastic constants at high temperature, and from the sample surface, which could apply additional stress on the loop, to explain the difference of internal stress.

To conclude, 1) the surface issue needs to be clarified and 2) the appropriate citation needs to be included. 5) Cannot be further resolved until potentials that are able to describe the elastic phenomenon become available. Until then, all high temperature MD work must be qualified with the fact that the elastic constants are poorly described.

Answer: Following your suggestions, we provided more information about image force calculation based on Cai's theory to prove that the present computational box is large enough and the image energy can be neglected. We also cited a paper from LANL group to confirm that the results obtained with SAAMD are same as those obtained with ParSplice method. The uncertainty from the empirical potential to describe the elastic constants at high temperature and its effect on internal stress difference between the simulations and experiments have been included in the revised manuscript. We do wish that these answers will address all the concerns of the reviewer. We would like to take this opportunity to thank you for your comments and suggestions, which allows us to further improve the quality of the manuscript.

Reviewer #3 (Remarks to the Author):

Contrary to previous assumptions, the manuscript by Gao et al predicts that a type of interstitial loop in BCC Fe is in fact mobile at room temperature. If true, this finding would represent a significant advance in our understanding of irradiated materials. However, the simulations are not entirely convincing. More detailed comments are provided below.

1). *First, I am satisfied that the system size used in this study is sufficiently large such that effects of neighboring defects from the periodic boundary conditions do not substantially affect the results. (However, I do agree that the cutoff of the potential is irrelevant in this discussion and that sentence in the paper – line 306 -should be removed.) Second, it is true that some EAM potentials do not capture the energetics of interstitials well, but the authors have tested several different potentials and apparently there are no obvious differences in the results.*

Answer: Thanks referee for these suggestions. Following your suggestion, we have removed the sentence related to the cutoff of the potential in the discussion section.

2). *The main concern I have – and it is a major one – was discussed (I think) in comment number 6 of the previous reviewers. The simulations show that the <100> loops can adopt different habit planes from thermal fluctuations. That would mean, however, that different habit planes should be observed in experiment. I admit I am not completely up to date on experimental findings, but it seems to me only the 100 habit has been found. If so, then this discrepancy is a serious problem and calls into question the entire simulation methodology.*

Answer: Thanks referee for this question. From experiments, it may be the first time to report that there exist different habit planes for <100> loops. Our TEM observations clearly confirmed the existence of these habit planes (please see the Movie2 and 3 in Supplementary Information). In order to confirm these habit planes, we further simulated the TEM images with the TEMACI program (<https://www.materials.ox.ac.uk/research/rippublications/temaci.html>) developed by Dr. Zhou in Oxford. This program can provide TEM amplitude contrast imaging from straight dislocations and dislocation loops based on solving numerically the Howie-Basinski equations. The details of this program can be found in Dr. Zhou's papers. (Z. Zhou, S. L. Dudarev, M. L. Jenkins and A. P. Sutton, *Inst. Physics Conf. Ser.* **179**, 203, 2003; Z. Zhou, S. L. Dudarev, M. L. Jenkins, A. P. Sutton, *MRS Proceedings* **792**, 491, 2004; Z. Zhou, A. P. Sutton, S. L. Dudarev, M. L. Jenkins and M. A. Kirk, *Proc. R. Soc. London A* **461**, 3935, 2005; Z. Zhou, M. L. Jenkins, S. L. Dudarev, A. P. Sutton and M. A. Kirk, *Phil. Mag.*, **86**, 4851, 2006). The comparison between the simulated TEM images with TEMACI for a <100> loop located on (100), (-310), (-210) and (-110) habit planes and *in-situ* TEM results shown in Movie-3 are exhibited in Fig. A1. In the TEMACI image simulations, the zone axis is taken as [100], diffraction vector is [011] and habit planes are set to be (100), (-310), (-210) and (-110). Accelerating voltage is 200 kV. The comparison clearly indicates that the habit planes observed *in-situ* experiment are same as the habit planes in the simulated TEM images, i.e. (100), (-310), (-210) and (-110) habit planes. Therefore, the loop shown in Movie-3 takes different habit planes during its migration. With these TEM simulation results, the properties of these habit planes have been confirmed, which are same as the SAAMD predictions.

Fig.A1 Comparison between the simulated TEM images and *in-situ* images for a $\langle 100 \rangle$ loop located on different habit planes. The top images are the results simulated by TEMACI program, while the bottom images are results taken from the *in-situ* observations from Movie-3.

In addition to these experimental results, in literatures, there were also theoretical calculations to predict the possibility of $\langle 100 \rangle$ loops with the different habit planes of $\{100\}$ and $\{110\}$. (A. B Sivak, V. A. Romanov and V. M. Chernov, AIP Conference Proceedings, 999, 118, 2008).

In our paper, it is for the first time to clearly observe the different habit planes of a $\langle 100 \rangle$ loop and quantitatively determine these planes. The change of these habit planes provides a driving force for a $\langle 100 \rangle$ dislocation loop to migrate, a new mechanism for the diffusion of a $\langle 100 \rangle$ loop. These different habit planes are also confirmed through the *in-situ* TEM observations (please see Fig. 2 and 3 in which the different habit planes of the loops are clearly indicated). We agree with you that these results would represent a significant advance in our understanding of irradiated materials.

The boost potential scheme originally proposed by Voter works well for cases such as surface diffusion, where every adatom site on the surface has the same potential energy landscape. In these cases it is straightforward to generate a boost that achieves acceleration, but does not alter the potential energy in the vicinity of the saddle point. In recent years MD techniques have been developed where the boost can be used for varying energy landscapes. The SSAMD is one such technique. In the original SSAMD paper the authors examined the case of He atom clustering. Here you would expect the energy landscape of He to vary depending on the presence of nearby He atoms, but the energy landscape, I would think, does not vary significantly. In the case of a PIDL I would expect the energy landscape to change substantially for atoms all along the edge of the loop. Therefore, how do we know the SSAMD is not introducing errors? Also, as an interstitial atom jumps out of the plane of the loop, then, again, I would assume the potential energy surface for that atom has now changed considerably. Is this accounted for in the SSAMD? Should it be?

Answer: Thanks the referee for this good question. In our current study, the diffusion of a $\langle 100 \rangle$ loop by adopting different habit planes cannot be described through SIA jumps one by one out from the original habit plane to form a new habit plane. Actually, the present results indicate that there is no dissociation and recombination process occurred during its diffusion, but instead, a small vibration amplitude of SIAs along the $\langle 100 \rangle$ direction can lead to the gradual change from one habit plane to another after enough time steps, for example, from (100) to (170), (150), (130), (120) and then (110) planes. The maximum amplitude of one atom between two continuous steps (one vibration) at 600 K is around 0.6\AA , and the typical vibration amplitude is around $0.2\text{-}0.3\text{\AA}$ for the present SAAMD simulations after applying the boost potential. Therefore, it is expected that the energy landscape does not change substantially for the atoms all along the edge of the loop.

The small vibration magnitudes would not result in the dissociation of an interstitial from a $\langle 100 \rangle$ loop as the binding energy E^b is more than 3.2 eV, as shown in the following figure (Fig. A2).

Fig. A2 Binding energy of an interstitial to a $\langle 100 \rangle$ loop containing N_{SIA} SIAs

From this figure, it is clear that it is very difficult for an SIA to dissociate from the $\langle 100 \rangle$ loops with E^b above 3.2 eV, which confirms the gradual rotation process to a different habit plane suggested in the present work.

Furthermore, according to the theory of potential energy surface, the Taylor's series about the minimum can be used to approximate the potential energy surface, which is a function of atomic coordinates in our work. Since the SAAMD method provides a gradual rotation process for a $\langle 100 \rangle$ loop to different habit planes, the initial system potential and related boost potential could keep their continuities not only for their functions, but also for their 1st derivatives and even the 2nd derivatives, which have been proved in our paper (Ref [15]). Therefore, based on above analysis, we could conclude that after applying boost energy, the potential energy surface explored by the SAAMD does not change significantly, and thus the results based on above simulations can be correctly compared to the *in-situ* TEM observations.

Therefore, SAAMD has fully considered the effect of PIDL on energy surface of the system. Based on above explanation, it is clear that in the case of a PIDL, the energy landscape would not change substantially for the atoms all along the edge of the loop. The continuous and gradual rotation process of habit planes ensures the correction of these simulations after applying the boost potential.

In the SSAMD two parameters are used to compute the (time varying) boost potential. One is a prefactor E_b and the second is a cutoff distance q such that the boost potential goes to zero when the cutoff is reached. The authors never state what E_b is or how it is obtained. They should. More important, the authors have used a maximum cutoff of $q=2A$. This value seems completely incorrect. The nearest neighbor interstitial distance in BCC iron is considerably less than $2A$ and therefore during an interstitial hop a bias potential is applied during the entire path?? Since the saddle point will be roughly half way along the path the value of q seems to be too large by a least a factor of two. I realize there is a word limit on Nature Comm papers, but this point needs to be explained in more detail. How did the authors choose E_b and max q ? What simulations were used to set these parameters? Can the same parameters be used for the PIDL case? How do we know? Perhaps the authors can re-run the simulation leading to movie 1, but vary the SSAMD parameters over a large range and

verify that the changing habit plane is not an artifact of the simulation method.

Answer: Thanks referee for these questions. Firstly, we would like to bring the referee's attention that the current SSAMD is slightly different from the boost potential scheme originally proposed by Voter works, where the boost potential **is slowly increased** to accelerate the dynamics. The current approach guaranties the continuity of the potential, its 1st derivatives and even the 2nd derivatives, and thus, the dynamics simulations are very stable. Turning to the question about E_b , we have described a method to determine the value of E_b and also ΔE_b in our previous paper. Following your suggestion, we also provided the related information in the supplementary materials. In our paper (Ref. [15]), E_b is defined as a self-adaptive parameter in the boost potential scheme, which is increased to $E_b + \Delta E_b$ at each accelerated step. In terms of q , please note that q is not a maximum distance cutoff for a jump along the path, but it is the total displacement of atoms in an active volume (AV) within the system initially, with an increment of $q + \Delta q$ at each accelerated step (Δq is a small value $\sim 1\%$ of initial total displacement). To make it clearer, we describe here the SAAMD method in detail. We hope that these explanations would be helpful for the reviewer to understand our simulations in the present work.

The boost potential takes the following form (Eq. A1), which could be obtained from the state of atoms on their harmonic vibration around their stable atomic positions, as we explained in our paper:

$$V_{bias}(\{r_1, \dots, r_{N_{AV}}\}; t) = E_b(t) \left\{ 1 - \left[\frac{\xi(\{r_1, \dots, r_{N_{AV}}\})}{q(t)} \right]^2 \right\} H[q(t) - \xi(\{r_1, \dots, r_{N_{AV}}\})], \quad (A1)$$

where E_b and q are parameters, as a function of simulation time t . ξ is the total displacement of atoms related to their vibrations in the active volume (AV) at a given temperature. Here, we defined AV as a cylinder around the <100> loop with a radius 20% larger than that of the loop. H is the Heavyside step-function to ensure when atoms in the AV are far away from equilibrium, the boost potential becomes zero. Therefore, the boost potential developed in this work provides a way to keep the system vibrating around the state determined by $q(t)$. The self-adaptive feature of SAAMD is that E_b and q evolve with time. The underlying physics for this self-adaptive is to make sure the system evolve from the local minimum state to the higher energy state gradually. The initial values of E_b and q start at small values, initially yielding a very modest bias. In this work, the E_b starts from 1 eV and q starts from $\xi_0 + 0.1 \text{ \AA}$, where ξ_0 is the initially total displacement of atoms due to their vibrations at a given temperature. For example, before applying the boost potential, the system after molecular static relaxation would be further relaxed at a given temperature for several thousand MD steps. ξ_0 is then determined by the total displacement of atoms in AV after above MD relaxations, averaging over many time steps. It should be noted that the large Δq and ΔE_b would result in the significant fluctuations of the transition time from one state to another, thus leading to poor statistics. Following the transition state theory and previous AMD algorithms, V_{bias} should be limited to within small value (e.g. less than 0.5 eV for around total 1700 atoms in activation volume in the present work, i.e. 0.00029 eV each atom) to void the significant change of energy surface but keep the high computational efficiency, according to our experience. Thus, Δq is generally taken around 1% of the initial total displacement ξ_0 , that is around 3.0 \AA in this work. According to equation A1, ΔE_b can be determined as around 2.5 eV to 3.0 eV. In this way, the increase of E_b can restrain the total displacement ξ close to $q(t)$ after applying the boost potential, to ensure the system changes smoothly. The stabilities of these parameters have also been tested by our SAAMD method, and the time fluctuation for each jump is shown in Fig.A3 with a Δq of 3.0 \AA .

Fig.A3 Time Fluctuation of each jump for a $\langle 100 \rangle$ loop diffusion with $\Delta q \sim 3.0 \text{ \AA}$ and $\Delta E_b \sim 2.5 \text{ eV}$.

Although the values of Δq and ΔE_b can be changed, for a given system, the ranges of these two parameters are limited. If we change them to smaller values, much more computational time is required for the system to evolve from the local state to saddle point. Following your suggestions, we performed more SAAMD simulations with small $\Delta q \sim 1.0 \text{ \AA}$ and $\Delta E_b \sim 1.0 \text{ eV}$ as shown in newly provided movies (movie-A1, movie-A2). In these simulations, the rotation between different habit planes has been confirmed, thus driving the $\langle 100 \rangle$ loop to diffuse, but it takes much more time (around one month) to complete simulation, generally 8 to 9 longer than the simulations with $\Delta q \sim 3.0 \text{ \AA}$ and $\Delta E_b \sim 2.5 \text{ eV}$. If we change them to larger values, either the over-boost or poor statistic of time would be obtained. The example simulations with $\Delta q \sim 8.0 \text{ \AA}$ and $\Delta E_b \sim 6.0 \text{ eV}$ results in the poor statistic of the time for jump as shown in Fig.A4.

Fig.A4 Time Fluctuation for each jump for a $\langle 100 \rangle$ loop diffusion with $\Delta q \sim 8.0 \text{ \AA}$ and $\Delta E_b \sim 6.0 \text{ eV}$.

The point about $q=2A$, as mentioned by reviewer, is in fact related to the dividing surface between two states, but not a maximum distance cutoff, as we explained above. Instead, it is the maximum displacement of an atom (D_{\max}) in the AV to determine the criteria to stop the boost. Therefore, when one atom has its displacement larger than D_{\max} , the boost potential will be removed and only the originally empirical potential is applied to describe the interaction between atoms in the further continue simulation. As shown in Fig. A5, when the boost is removed, the continuous dynamic simulation would result in the system overcoming the small energy barrier at the saddle point and thus, move to another local state. However, D_{\max} is not necessary to set exactly same as the saddle point, but generally less than the saddle point state.

[REDACTED]

Fig. A5 Schematic of SAAMD algorithm adapted from Ref[15]

In this work, we focus on a $\langle 100 \rangle$ loop which is formed by $\langle 100 \rangle$ interstitials along its Burgers vector. Following your idea, we measured the nearest neighbor interstitial distance of one $\langle 100 \rangle$ interstitial itself and that in a $\langle 100 \rangle$ loop in BCC iron, which is around 2.04 Å and 2.26 Å, respectively, after full relaxation. The value of the Burgers vector of a $\langle 100 \rangle$ loop is around 2.8 Å. All these values are larger than 2 Å.

Furthermore, it should be noted that a $\langle 100 \rangle$ loop diffuses from the initial $\{100\}$ habit plane along its Burger vector to another habit plane, rather than a single $\langle 100 \rangle$ dumbbell to diffuse along the $\langle 100 \rangle$ direction. In our paper, we have described that if only the classical MD was performed, a $\langle 100 \rangle$ loop is sessile and vibrates with a small amplitude around its equilibrium position. If we carefully follow the diffusion of one of the $\langle 100 \rangle$ crowdions in a $\langle 100 \rangle$ loop, we are able to understand the underlying physics. As shown in Fig. A6, when a $\langle 100 \rangle$ loop initiates its rotation to other habit plane, it needs to move the mass center of its $\langle 100 \rangle$ crowdion away from the original site. If this movement is limited, the loop remains, because the $\langle 100 \rangle$ crowdions would move forward and backward from its original site, without leading to the diffusion of the loop. Thus, as shown in Fig. A6, for the rotation of a $[100]$ loop from its original habit plane (100) to (-310) , the mass center of **one $[100]$ dumbbell** should move from R1 region (close to atoms a and b) to R2 region (close to atom 4 and 5), which requires the movement of atom a to 1, b to 2, c to 3, d to 4, e to 5 and f to 6 (please note that each atom moves a small distance). Here, atoms a to f are original sites and 1 to 6 are new sites after the crowdion movements. In order to reach such a movement, the maximum distance from atom c to 3 should be satisfied. This value is around 2.4 Å. Considering the effect of thermal fluctuation during the diffusion of loop, the D_{max} should be less than 2.4 Å. We then did the pre-simulation from $D_{max} = 2.4$ Å and decreased it gradually to test whether the system could overcome the saddle point state by SAAMD simulations. After enough pre-simulations, we determined that the value of D_{max} around 2 Å is large enough to trigger the rotation.

Fig. A6: Schematic to determine the theoretical maximum displacement D_{max}

3). I do not understand the discussion surrounding eq. 1. The authors state the MD simulation was modified using this equation. How exactly is the equation used? How is a jump frequency incorporated into a MD simulation? What is the nature of the stress applied? Is it a hydrostatic pressure? Uniaxial tension? What is the energy barrier in this equation? Is it the energy barrier between the PILD on different habit planes? Finally what is the prefactor?

Answer: Sorry for confusing. In this work, equation (1) is in fact an Arrhenius equation, which is only used to qualitatively indicate that the energy barrier is a function of applied stress and is not incorporated into a MD simulation. In terms of the stress applied, we followed the same method developed by Feichtinger, Derlet and Van Swygenhoven (PRB, 67, 024113, 2003) to perform the nano-indentation simulation, but with a modification to relax the system after each time of indentation (relaxation for 100 ps for around 0.1 Å indentation). When this method is used, indentation force is a function of the indentation depth. Both of them are variables. With this method, the diffusion of the $\langle 100 \rangle$ loops could be induced by the external stress created by the indentation, even for a larger loop. Following its diffusion process, the rotation of habit plane is also observed as a main mechanism for its diffusion, which is the same as the results obtained by SAAMD method. Therefore, two computational approaches can be used to validate each other. In order to avoid the misleading, we removed this equation in the revised manuscript.

4). On line 150 the authors state "... the $\frac{1}{2}\langle 111 \rangle$ loop intersects at a line, whereas the $\langle \text{loop} \rangle$ intersects at a point." What does "intersect" mean here? What is intersecting with what?

Answer: Thanks the referee. Sorry that we have not made this clear in our manuscript. Here the intersect is used to describe the geometric arrangement of the habit planes of a $\langle 100 \rangle$ loop or a $\frac{1}{2}\langle 111 \rangle$ loop. For example, when a $\frac{1}{2}\langle 111 \rangle$ loop diffuses, its habit planes generally include the (111), (112) and (110) planes. All these three planes have the same rotation axis, and thus, this axis is defined to be their geometric intersect. For a $\langle 100 \rangle$ loop, its habit planes include the (100), (120), (110) and (102) and (101), as shown in Fig.2 in the paper. Geometrically, these planes intersect at a point.

REVIEWERS' COMMENTS

Reviewer #3 (Remarks to the Author):

I am satisfied with the revised manuscript and the more detailed explanations. I congratulate the authors for their nice work.

REVIEWER COMMENTS

Reviewer #3 (Remarks to the Author):

I am satisfied with the revised manuscript and the more detailed explanations. I congratulate the authors for their nice work.

We would like to take this opportunity to thank you for your comments and suggestions, which allow us to further improve the quality of the manuscript.